# NCKAP1 Inhibits the Progression of Renal Carcinoma via Modulating Immune Responses and the PI3K/AKT/mTOR Signaling Pathway

**DOI:** 10.3390/ijms26062813

**Published:** 2025-03-20

**Authors:** Xin Zhang, Jianqing Ye, Lixiang Sun, Wanli Xu, Xiaomeng He, Juan Bao, Jin Wang

**Affiliations:** 1Central Laboratory, Zhongshan Hospital (Xiamen), Fudan University, Xiamen 361015, China; 23111300002@m.fudan.edu.cn (X.Z.); sun.lixiang@zsxmhospital.com (L.S.); xu.wanli@zsxmhospital.com (W.X.); 2Department of Urology, Xinhua Hospital, School of Medicine, Shanghai Jiaotong University, 1665 Kongjiang Road, Shanghai 200092, China; ye910@126.com; 3Shanghai Public Health Clinical Center, Fudan University, 2901 Caolang Road, Jinshan District, Shanghai 201508, Chinabj901120@163.com (J.B.)

**Keywords:** NCKAP1, immunotherapy, pancancer, renal cancer

## Abstract

Nck-associated protein 1 (NCKAP1) is critical for cytoskeletal functions and various cellular activities, and deregulation of NCKAP1 in many cancers significantly influences the outcomes of malignant diseases. However, the functions of NCKAP1 in the progression of renal cancer are yet unknown. To investigate the specific roles of NCKAP1 in the immune regulation and tumor progression of renal cancer, the expression of NCKAP1 and genetic variations were analyzed across cancer types at different pathological stages via UALCAN and cBioPortal. Immune cell infiltration in renal cancer was also assessed by ssGSEA and single-cell gene expression data from the GEO. RNA sequencing of NCKAP1-overexpressing 769P cells further examined the impact of NCKAP1 on kidney cancer. Our pancancer analyses revealed a complex NCKAP1 expression profile across various cancer types, with reduced levels in renal cancer patients linked to patient prognosis. CIBERSORT and single-cell RNA sequencing revealed the expression patterns of NCKAP1 in different cell lineages in renal cancer and a significant correlation between NCKAP1 and immune cell infiltration in the kidney tumor microenvironment. We further verified that NCKAP1 suppressed cancer cell growth and affected tumor development in renal cancer via the PI3K/AKT/mTOR signaling pathway. Our results indicate that NCKAP1 is a potential predictive marker and treatment target for renal cancer.

## 1. Introduction

Renal cancer is the seventh most prevalent malignancy in both sexes, accounting for 4.08% of new cancer cases, and 81,610 individuals are projected to be diagnosed in the United States by 2024 [1]. Renal cell carcinoma (RCC) accounts for more than 90% of all renal epithelial cancers, with an associated mortality rate of approximately 20% [2,3]. Although marked advances in the treatment of RCC have been made over the past few decades, understanding these metabolic peculiarities and the roles of the immune microenvironment (TME) remains crucial for developing targeted cancer therapies [4]. However, no clear relationship among the degree of immunosuppression, cancer metabolism, and the risk of renal cancer has been defined [5], and cancer metabolism is characterized by unique metabolic behaviors in cancer cells, such as altered glucose uptake and energy production pathways, highlighting the importance of regulated cell death (RCD) in cancer metabolic therapy, especially through a novel form called disulfidocytosis, which is linked to redox state changes under glucose scarcity and can be triggered by increased levels of the SLC7A11 transporter, leading to cancer cell death by disrupting cytoskeletal integrity and emphasizing its therapeutic potential [6,7]. Disulfidptosis plays key roles in the progression of renal clear cell carcinoma (KIRC) through the immune microenvironment, and extensive interactions between disulfidptosis-mediated TME subgroups and tumor epithelial cells were discovered in renal cancer patients via CellChat analysis [8].

NCK-associated protein 1 (NCKAP1), an integral component of the WAVE regulatory complex, can activate the Arp2/3 complex to promote actin polymerization, and its absence hinders disulfidptosis, promotes cell survival, and potentially aids in tumor progression and treatment resistance [7]. NCKAP1 encodes a protein that is involved in cellular adhesion, cytoskeletal organization, and signal transduction and is associated with the pathogenesis of various neoplasms [9]. The crucial role of NCKAP1 in balancing cell death and survival underscores its importance in tumor biology and its involvement in breast cancer metastasis as well as in colon and lung cancers, making it a promising target for cancer therapy [9,10,11,12,13]. Lower levels of NCKAP1 mRNA and protein in ACNA cells correlate with increased growth and colony formation [14], whereas NCKAP1 silencing disrupts the WASF3 complex and inhibits breast cancer progression [10,15]. Furthermore, decreased NCKAP1 expression in individuals with hepatocellular carcinoma is correlated with unfavorable prognosis and adverse patient outcomes in clear cell renal cell carcinoma (ccRCC) [16]. In summary, NCKAP1 has been linked to various cancers as a biomarker and has key roles in multiple cancers. However, its mechanisms and clinical implications in renal cancer remain limited.

Here, we focused on renal cancer to elucidate its specific functions and underlying mechanisms of immune regulation and tumor progression by comprehensively investigating the roles of NCKAP1 across cancers via extensive multiomics analysis of NCKAP1 across various cancers and highlighting its role in ccRCC tumorigenesis. Our findings increase our understanding of the clinical importance of NCKAP1 and its potential as a therapeutic target. This comprehensive single-gene study has substantially advanced our understanding of the molecular functions of NCKAP1 and provided crucial insights into the mechanisms of renal cancer development.

## 2. Results

### 2.1. Analyses of NCKAP1 Expression and Genetic Alterations Across Tumors

We initially evaluated NCKAP1 expression levels in both malignant and healthy tissues from 34 different cancer types in the TCGA datasets. We demonstrated elevated expression levels of NCKAP1 in multiple cancer types, including cholangiocarcinoma (CHOL), head and neck squamous cell carcinoma (HNSC), liver hepatocellular carcinoma (LIHC), and stomach adenocarcinoma (STAD). In contrast, reduced NCKAP1 expression in renal clear cell carcinoma (KIRC), kidney renal papillary cell carcinoma (KIRP), kidney chromophobe renal cell carcinoma (KICH), and prostate adenocarcinoma (PRAD) was also identified (Figure 1A and Appendix A). Differential NCKAP1 was also revealed via the GEPIA platform across different cancer stages, especially for bladder urothelial carcinoma (BLCA), KIRC, and several other specific cancer types, with noteworthy outcomes (Figure 1B). In parallel, a marked reduction in NCKAP1 protein expression was verified by UALCAN, which suggests a potential discrepancy between NCKAP1 mRNA and protein levels in various cancers, highlighting the complexity of its regulatory mechanisms. Suppression of gene activity was particularly evident in cancers originating in the breast, colon, lung, and kidney tissues (Figure 1C). The HPA database provided evidence of both the cytoplasmic and nuclear distributions of NCKAP1 (Figure 1D). Moreover, NCKAP1 genetic variations were investigated via cBioPortal, which indicates that endometrial cancer (EC) has the highest incidence of alterations, with approximately 6% of the cases affected. Among these alterations, mutations were the most prevalent (Figure 1E). We found that copy number alterations (CNAs) were prevalent in several cancers, with melanoma showing an approximately 4% frequency and bladder cancer, colorectal cancer (CRC), and mature B-cell neoplasms showing an approximately 3% frequency. The prevalence of NCKAP1 amplification was highest in ovarian epithelial tumors, occurring in approximately 2.5% of cases. In comparison, this genetic alteration was observed less frequently in non-small cell lung cancer (NSLC) and head and neck cancer, with an incidence of approximately 1.5%. Among them, prostate cancer had the highest percentage of “deep deletions”. Interestingly, only head and neck cancer and NSLC exhibited a minor percentage of “multiple alterations” that were absent in other cancers. In addition, the status of NCKAP1 in various cancers, including missense, truncating, in-frame, splice, and fusion mutations, is illustrated. Notably, the R343H mutation in the peptidoglycan-binding domain was the most common mutation, appearing in one patient each for CRC, EC, and NSLC (Figure 1F). This comprehensive analysis highlights the diverse genetic landscape of NCKAP1 across various cancers, emphasizing its potential impact on cancer pathophysiology and its therapeutic targets.

### 2.2. CIBERSORT and Single-Cell Transcriptomics Revealed a Significant Correlation Between NCKAP1 and Immune Cell Infiltration in the Kidney Tumor Microenvironment

To elucidate the critical importance of the tumor immune microenvironment in the initiation and progression of kidney malignancies, we investigated the relationship between NCKAP1 expression and the TME and demonstrated that the infiltration levels of 24 immune cell types in cancer were correlated with NCKAP1 expression via lollipop graph analysis (Figure 2A), which revealed a notable positive association between NCKAP1 and several immune cell types, including Tcm cells, neutrophils, mast cells, eosinophils, and Tgd cells. Conversely, NCKAP1 was negatively correlated with NK CD56bright cells, cytotoxic cells, and regulatory T (Treg) cells. Furthermore, we employed the CIBERSORT algorithm to assess the prevalence of 22 immune cell subtypes in patient cohorts with high and low NCKAP1 expression, with statistical significance determined by a *p* value less than 0.05. The infiltration rates of these immune cell populations are shown in Figure 2B. A boxplot illustrating the comparative infiltration levels of 24 immune cells between the high and low NCKAP1 groups is shown in Figure 2C. The analysis revealed that the low-NCKAP1 group presented elevated levels of CD8^+^ T cells, cytotoxic T cells, NK CD56 bright cells, Treg cells, and total T cells. Conversely, the high-NCKAP1 group presented increased levels of eosinophils, mast cells, neutrophils, NK cells, Th cells, central memory T cells (Tcm), Tem, Tgd, and Th17 cells. To gain deeper insight into the relationship between NCKAP1 and immune cells, coexpression analysis was performed, which revealed significant correlations. NCKAP1 exhibited a significant negative correlation with CD56 bright cells, cytotoxic T cells, and Treg cells. In contrast, a statistically significant positive correlation was observed with Tcm cells, neutrophils, mast cells, eosinophils, and Tgd cells (Figure 2D). Subsequent analysis revealed significant correlations between NCKAP1 expression and the 24 types of immune cells (Figure 2E). Our findings underscore the intricate relationship between NCKAP1 expression and immune cell dynamics in the TME, suggesting that NCKAP1 plays a role in modulating tumor behavior and immune responses in cancer.

Furthermore, we applied comprehensive scRNA-seq to investigate the function of NCKAP1 in the renal cancer microenvironment via data from the TCGA database. Six RCC samples were subjected to single-cell transcriptional analysis. Using the UMAP algorithm for cell clustering analysis, we classified the cells into 17 and 8 distinct clusters and manually annotated them as follows: T cells, CD8^+^ cytotoxic T lymphocytes (CTLs), proximal T cells, NK cells, macrophages, pericytes, Treg cells, monocytes, proliferating T cells, mature B cells, endothelial cells, myeloid dendritic cells (mDCs), mast cells, tumor cells, memory B cells (memB cells), myeloid dendritic cells, and fibroblasts (Figure 2F). The distribution of NCKAP1 in the UMAP plot of the renal cancer single-cell sequencing data was subsequently examined (Figure 2G). Next, we compared NCKAP1 expression among the various cell types identified in KIRC (Figure 2H), which indicated that NCKAP1 expression was not uniform across all cell identities, with certain cell types exhibiting significantly higher expression levels. For example, cells categorized as “pericytes” or “fibroblasts” presented elevated expression levels, as indicated by the taller bars in the chart, which suggests that NCKAP1 may play a significant role in the functions or differentiation processes specific to these cell types. The enhanced expression in pericytes could be associated with their role in maintaining vascular integrity and regulating blood flow, whereas, in fibroblasts, it might be linked to their involvement in extracellular matrix production and wound healing processes. Conversely, minimal to no expression is observed in cell types such as “mDC”, “men B-cell”, and “myeloid dendritic”, as shown by the absence or very short bars in the chart. Our findings suggest that NCKAP1 is not a critical factor in the normal physiological functions of these immune cells or that its expression is tightly regulated and may be induced only under specific pathological conditions. The differential expression of NCKAP1 across these cell types underscores the potential for NCKAP1 to be involved in diverse biological processes and could have implications for understanding the molecular mechanisms underlying renal cancer. Further investigations into the functional significance of NCKAP1 in these cell types may provide valuable insights into its role in renal cancer.

### 2.3. Reduced NCKAP1 Expression Was Correlated with More Advanced Stages of Pathological T Stage, Pathological Stage, and Higher Histological Grade and Significantly Affected the Overall Survival of Patients with Renal Cancer

Next, we assessed the prognostic significance of NCKAP1 in diverse cancers from the TCGA dataset via the GEPIA2 tool. Our analysis, presented through heatmap visualization, focused on the correlation between NCKAP1 expression and survival metrics, OS and PFS, to evaluate its potential as a prognostic biomarker (Figure 3A). The color gradient from blue to red represents the hazard ratio (HR), with blue indicating lower expression levels and red indicating higher expression levels. In certain tumor types, higher expression levels (red) are associated with a poorer prognosis, whereas, in others, lower expression levels (blue) correlate with a worse outcome. Specifically, high NCKAP1 expression correlated with poor survival in adrenocortical carcinoma (ACC, *p* = 0.061), cervical squamous cell carcinoma (CESC, *p* = 0.020), and LIHC (*p* = 0.037) patients, indicating that NCKAP1 may serve as an adverse prognostic factor in those cancer types. In contrast, high NCKAP1 expression was associated with better prognosis in KIRC (*p* = 0.004) and testicular germ cell tumors (TGCTs, *p* = 0.038). These findings highlight the role of NCKAP1 in cancer progression and its value as a survival biomarker that could guide treatment and patient management. Analysis of PFS revealed that high NCKAP1 expression in KIRC (*p* = 0.0038) and LGG (*p* = 0.018) correlated with improved prognosis, whereas high NCKAP1 expression in ACC (*p* = 0.005) and uveal melanoma (UVM, *p* = 0.032) was associated with poorer outcomes, which emphasized the importance of NCKAP1 as a biomarker for cancer survival and its impact on patient outcomes, suggesting its potential for personalized therapeutic approaches on the basis of gene expression profiles. To further investigate the role of NCKAP1 in renal cancer, we employed Kaplan–Meier (KM) survival curves to assess OS and disease-free survival (DFS) and specifically examined its influence on KIRC (Figure 3B,C). Our results revealed that the blue line (low NCKAP1) was associated with a lower OS percentage and a lower DFS percentage than the red line (high NCKAP1). High NCKAP1 expression is correlated with better survival, suggesting its potential as a prognostic biomarker. Furthermore, the forest plot demonstrated that increased expression of NCKAP1 was significantly associated with improved OS in patients with KIRC (HR = 0.504, 95% CI: 0.415–0.766, *p* = 0.0002). Elevated levels of NCKAP1 in KIRC are associated with a better prognosis, indicating a positive prognostic significance.

To evaluate the clinical significance of NCKAP1, we analyzed the relationships between NCKAP1 expression and clinical features in 897 patients with renal cancer, including KIRC (*n* = 541), KIRP (*n* = 291) and KICH (*n* = 65) from the TCGA datasets (Table 1). Reduced NCKAP1 expression in KIRC was significantly correlated with more advanced stages of pathological T stage (*p* = 0.035), overall pathological stage (*p* = 0.006), and higher histological grade (*p* = 0.0004). We also demonstrated that NCKAP1 expression significantly affected OS (*p* < 0.0001), DSS (*p* < 0.0001), and PFI (*p* = 0.002) in patients with KIRC, although NCKAP1 expression was not significantly correlated with pathological grade or clinical prognosis in KIRP or KICH patients (Table 1 and Appendix A). Moreover, the relationships between NCKAP1 mRNA expression and various clinicopathological features were assessed (Figure 3D). The expression of NCKAP1 mRNA was significantly correlated with several factors, including advanced pathological T and M grades, tumor grade, disease stage, and sex. Notably, stage 4 and grade 4 tumors presented markedly reduced NCKAP1 expression. Importantly, high expression of NCKAP1 was correlated with a better prognosis. Our findings underscore the significant impact of NCKAP1 on the clinical outcomes of kidney cancer patients, highlighting its potential as a prognostic biomarker and a promising target for therapeutic interventions.

### 2.4. NCKAP1 Inhibits the Proliferation and Invasion of Renal Cancer Cells by the PI3K/AKT/mTOR Signaling Pathway

To explore the role of NCKAP1 in kidney cancer, the effects of NCKAP1 on key cellular functions, such as proliferation, migration, and invasion, were evaluated (Figure 4). First, NCKAP1 expression levels were compared between paracancerous and tumor tissues, revealing a significant reduction in NCKAP1 expression in tumor tissues, suggesting a potential role for NCKAP1 in tumorigenesis (Figure 4A,B). IHC assessment of kidney tissues from the Human Protein Atlas corroborated these findings, demonstrating that NCKAP1 protein expression was lower in renal cancer tissues than in normal kidney tissues (Figure 4C). Moreover, mRNA analysis of renal carcinoma cells revealed that the expression level of NCKAP1 in multiple renal cancer cell lines was significantly lower than that in the normal cell line HK-2 (Figure 4D). We subsequently established knockdown and overexpression models of NCKAP1 in 786O and 769P cells and validated the expression of NCKAP1 at both the RNA and protein levels (Figure 4E,F). To further elucidate the role of NCKAP1 in renal cancer, we conducted a series of functional assays to validate its functional contributions. Initially, NCKAP1 expression was evaluated in various renal cell lines and a normal kidney cell line. On the basis of these results, 769P and 786O cells were chosen for further investigation. The CCK8 assay, as shown in Figure 4G, was used to assess and quantify the rate of cellular proliferation. Compared with NCKAP1-overexpressing cells, NCKAP1-knockdown cells presented increased proliferation rates, indicating the inhibitory role of NCKAP1 in the proliferation of renal cancer cells. Transwell migration and Matrigel invasion assays revealed that cells with downregulated NCKAP1 expression presented increased migratory and invasive capabilities (Figure 4H), suggesting that NCKAP1 suppresses the metastatic potential of renal cancer cells. Collectively, these findings suggest that alterations in NCKAP1 expression significantly influence the aggressive characteristics of renal cancer cells. These effects include effects on proliferative capacity, migratory potential, and invasive properties. These findings underscore the potential of NCKAP1 as a therapeutic target for the management of renal cancer.

To explore the underlying mechanisms of the effect of NCKAP1 on kidney cancer, we conducted RNA sequencing analysis of 769P cells with increased NCKAP1 expression. Volcano plot analysis revealed a substantial number of differentially expressed genes (DEGs) in NCKAP1-overexpressing cells compared with control cells, with statistical significance and substantial fold changes (FCs) (Figure 5A), indicating the broad regulatory effects of NCKAP1. This analysis identified 729 upregulated and 136 downregulated genes, defined by differential expression with a log2-fold change threshold of ±1 and a *p* value < 0.05. The differential expression analysis of NCKAP1 highlighted the 30 genes whose expression was most significantly upregulated or downregulated, the results of which are presented in Appendix A. The heatmap in Figure 5B visually represents the transcriptional profile changes, with a clear distinction between upregulated and downregulated genes. Cancer-pathway-related KEGG enrichment analysis (Figure 5C) and KEGG pathway (Figure 5D and Appendix A) analyses of the top 30 upregulated and downregulated genes revealed the roles of NCKAP1 in the MAPK signaling pathway influenced by NCKAP1. Moreover, immunoblotting was also used to investigate the regulatory effect of NCKAP1 on the PI3K/AKT/mTOR signaling pathway in renal cancer cells (Figure 5E). Taken together, our results indicate that NCKAP1 is crucial for the biological processes of renal cancer cells and potentially acts as an essential regulator of cell growth, movement, and tissue penetration.

## 3. Discussion

The increased expression of NCKAP1 in kidney cancer indicates its potential as a prognostic indicator for suppressing cellular proliferation in clear cell RCC [14]. NCKAP1 can serve as both an immune marker and a prognostic indicator across diverse cancer types [17,18] and significantly inhibits cancer cell migration and invasion as a novel prognostic marker in CRC [19]. Similar results were observed in gastric cancer, where NCKAP1 emerged as a promising prognostic biomarker correlated with actin dynamics, GTPase energy metabolism, immune infiltration, and immunotherapy [20]. In melanoma, NCKAP1 inhibition can slow tumor proliferation and growth, highlighting its role in cell cycle progression. NCKAP1 is also deregulated in renal cancer and is associated with patient outcomes [14,21]. Moreover, NCKAP1-depleted tumors exhibit fibrotic stroma with increased collagen deposition and enhanced immune infiltration, indicating that NCKAP1 plays a crucial role in the advancement of tumors and preservation of tumor tissue structure in kidney cancer. Therefore, NCKAP1 could be considered a cancer-associated gene with significant potential as a prognostic indicator in various types of cancers.

In this study, we investigated whether NCKAP1 was expressed at lower levels in renal cancer, which was correlated with more favorable clinical outcomes. Our findings indicate that NCKAP1 may function as a tumor suppressor in kidney cancer, highlighting its role in impeding tumor progression. To explore the potential molecular mechanisms underlying the role of NCKAP1 in tumorigenesis, we conducted a comprehensive analysis involving the identification of NCKAP1-interacting genes and NCKAP1-correlated genes via the tools STRING and GEPIA2, respectively. The resulting gene sets were subjected to pathway and process enrichment analyses to elucidate their functional implications. Appendix A shows 50 experimentally verified NCKAP1-interacting genes within the PPI network. Additionally, the top 100 NCKAP1-correlated genes were identified, with the six most significantly correlated genes, CAMSAP1, RAB3GAP1, SCRN3, SP3, SPTLC1, and ZFP91, highlighted on the basis of their high correlation coefficients. These correlations are visually represented in a heatmap showing a positive correlation between NCKAP1 and these six genes across a wide range of TCGA cancer types (Appendix A). Intersection analysis of NCKAP1-interacting and NCKAP1-correlated genes revealed three common genes: Abl Interactor 1 (ABI1), Mitogen-Activated Protein Kinase 1 (MAPK1), and Wiskott–Aldrich Syndrome Like (WASL), which suggests a functional relationship within cellular processes. ABI1, which is associated with actin cytoskeleton regulation, may collaborate with NCKAP1 in regulating cellular dynamics. MAPK1, a central player in the MAPK/ERK pathway, likely modulates growth and differentiation in conjunction with NCKAP1, impacting tissue homeostasis and disease progression. WASL, which is critical for actin polymerization and cell movement, may synergize with NCKAP1 to regulate cytoskeletal rearrangements. These interactions suggest that a complex network influences cell behavior, warranting further exploration to uncover the underlying mechanisms and their biological significance (Appendix A). We also performed GO and KEGG pathway analyses to examine DEGs linked to NCKAP1 in kidney cancer, the results of which are shown in Appendix A. Notably, NCKAP1 is implicated in “keratinocyte differentiation” and “keratinization”, which are critical for skin development and structure, as well as in the “acute-phase response” associated with inflammation. The gene is also related to “intermediate filaments” and “keratin filaments”, which are components of the cytoskeleton, suggesting a role in maintaining cell structure. To identify the functional pathways and biological processes associated with NCKAP1, we also conducted GSEA analyses of differentially expressed NCKAP1 genes in three types of renal cancer. The GSEA results for NCKAP1-associated DEGs in KICH and KIRC are presented in Appendix A, respectively. The GSEA results for NCKAP1-related genes in KICH indicated significant enrichment in pathways associated with calcium mobilization and B-cell receptor signaling, which are crucial for cellular communication and the immune response. In the context of KICH, these pathways may play a role in tumor progression and immune evasion, suggesting potential targets for therapeutic intervention (Appendix A). Conversely, in KIRC, the GSEA highlighted pathways such as the scavenging of heme from plasma and CD22-mediated BCR regulation, suggesting a role in heme metabolism and B-cell signaling, which could be linked to the immune response and progression of KIRC (Appendix A). These pathways may influence the tumor microenvironment and immune response, potentially affecting the progression and immunotherapy response of KIRC, offering new insights into the molecular underpinnings of RCC and possible avenues for targeted therapies [22,23,24]. Therefore, our study examined the interplay between NCKAP1 and immune system components and revealed positive correlations with Tcm cells, neutrophils, mast cells, eosinophils, and Tgd cells and negative correlations with NK CD56dim cells, cytotoxic T cells, and Tregs. Single-cell sequencing analysis revealed that NCKAP1 was more highly expressed in fibroblasts than in 17 other immune cell types. Additionally, we analyzed the downstream pathways regulated by NCKAP1 in renal cancer. NCKAP1’s regulation of the PI3K/AKT signaling axis highlights its crucial role in cell survival and proliferation. We verified that NCKAP1 suppressed the proliferation and invasion of renal cancer cells via the PI3K/AKT/mTOR signaling pathway. These observations are vital because they expand our knowledge of the diverse functions of NCKAP1 in cancer biology and emphasize its importance as a potential therapeutic avenue. The dual role of NCKAP1 as a tumor suppressor in some contexts and potentially promoting oncogenesis in others suggests that targeted therapies modulating its activity could provide significant benefits tailored to specific cancer types. Overall, our findings underscore the importance of NCKAP1-related genes in the pathophysiology of renal cancers and suggest that further investigations into these signaling pathways could reveal novel therapeutic targets and enhance our understanding of renal cancer biology.

## 4. Conclusions

In this study, we demonstrated that NCKAP1 inhibited the growth and progression of renal cancer cells via the PI3K/AKT/mTOR signaling pathway, providing a molecular basis for its potential as a therapeutic target. Furthermore, CIBERSORT and single-cell RNA sequencing data revealed that NCKAP1 had significant immunomodulatory effects. Considering the complexities revealed by our pancancer analysis, future research should focus on elucidating the mechanistic role of NCKAP1 in various cancers. Additionally, more clinical data, particularly from online databases and animal experiments, should be used to validate these findings. We should also investigate the specific pathways influenced by NCKAP1 and evaluate the impact of its modulation on clinical outcomes. These studies could pave the way for personalized therapeutic strategies, particularly for renal cancer, where the impact of NCKAP1 is more pronounced. This study advances our understanding of the roles of NCKAP1 in cancer and opens new avenues for personalized treatment strategies, potentially transforming clinical practice.

## 5. Materials and Methods

### 5.1. Clinical Characteristics and Genetic Alteration Analyses

We analyzed the protein expression of NCKAP1 in both cancerous and healthy tissues across various cancer types. The individual cancer stages in TCGA database (https://portal.gdc.cancer.gov) were applied to perform a pathological stage analysis for all TCGA cancers [25], generating box plots depicting NCKAP1 expression. The data were accessed on a date within June 2024. To conduct an in-depth analysis of multidimensional cancer genomic data, we analyzed TCGA Pan-Cancer Atlas Studies—Cancer Types within cBioPortal (www.cbioportal.org) to retrieve comprehensive information on the genetic alterations of NCKAP1 [26,27], including comprehensive information on the frequency of alterations, types of mutations, and copy number variations (CNAs) across all cancer types within the TCGA dataset. We also identified the specific mutated sites associated with NCKAP1. Additionally, we compared various survival metrics, such as overall survival (OS), disease-free survival (DFS), disease-specific survival (DSS), and progression-free survival (PFS), between patients with cervical and bladder cancer who had NCKAP1 genetic alterations and those who did not. The statistical methodology employed in this study incorporated the construction of Kaplan–Meier curves (KMs) and the computation of log-rank *p* values, thereby providing rigorous statistical support.

### 5.2. Survival and Similar Gene Analysis

To explore the prognostic correlation of NCKAP1 across all TCGA tumors, we applied survival maps and survival analyses to examine RNA sequencing expression information from the TCGA and GTEx databases. We conducted a comprehensive prognostic analysis via KM curves. The top 100 genes related to NCKAP1 were identified in both TCGA tumor tissues and corresponding normal tissues via expression analysis, and similar genes were detected via GEPIA2 (http://gepia2.cancer-pku.cn/#index, accessed on 25 January 2024) [25].

### 5.3. Gene Enrichment Analysis

To comprehensively analyze the genes associated with NCKAP1, we initially created a list of genes by combining the top 100 genes correlated with NCKAP1, as identified by GEPIA2, and the 50 genes known to interact with NCKAP1 through experimental validation, as reported in STRING. The combined gene list was input into Metascape (http://metascape.org, accessed on 30 January 2024) for further analysis [28]. After completion of the analysis, the gene list analysis report generated by Metascape was exported. The functional pathways and biological processes associated with NCKAP1-related genes were analyzed by GO enrichment and KEGG pathways.

### 5.4. Human Proteome Atlas and Protein–Protein Interaction Network Analysis

To gather extensive data on protein distribution across human tissues and cells, we utilized the Human Proteome Atlas (HPA) database (https://www.proteinatlas.org/, accessed on 10 March 2024) for further information. We downloaded immunohistochemical (IHC) images of NCKAP1 in kidney and renal cancer tissues. We also downloaded immunofluorescence images of NCKAP1 in tumor cells from the database. These images enabled us to visually analyze the subcellular distribution pattern of NCKAP1 in tumor cells. To analyze the interactions between coexpressed genes of NCKAP1, we utilized version 11.0 of the STRING database (https://string-db.org/, accessed on 17 March 2024) to create a protein–protein interaction (PPI) network [29]. We established the PPI network by setting the interaction score threshold to 0.4 and limiting the maximum number of interactors to 50 and identified genes that directly interact with NCKAP1 as central hub genes.

### 5.5. Single-Cell Transcriptomic Sequencing Data Analysis

From the GEO GSE178481 dataset, we obtained an expression matrix for ccRCC and normal epithelial cells on 9 September 2024. The data were converted to FASTQ format via the SRA Toolkit, followed by quality assessment via FastQC to evaluate the integrity of the sequencing reads. Low-quality sequences and adapters were removed via Trimmomatic software (version 0.39). The R package Giotto was used to process and visualize the data [16]. The normalized Giotto function was used to normalize the ccRCC count matrix. Highly variable genes were identified via the calculated HVF functions. Principal component analysis (PCA) was subsequently conducted on these genes via runPCA, and the JackStrawPlot function was subsequently employed to analyze the outcomes. Cell clustering was performed via the createNearestNetwork and doLeidenCluster functions, with a total of 17 cellular subtypes considered. Data visualization was performed via uniform manifold approximation and projection (UMAP) through the implementation of the RunUMAP and dimPlot2D functions.

### 5.6. Immune Cell Infiltration and Correlation Analysis Between Immune Cells and NCKAP1

A detailed analysis of 24 specific immune cells was conducted on 6 June 2024 to evaluate immune infiltration levels in renal cancer. Single-sample gene set enrichment analysis (ssGSEA) was conducted via the GSVA package in R to perform enrichment analysis of these cells. Spearman’s correlation analysis was used to evaluate the associations between NCKAP1 expression and immune cells. The assessment included various immune cell types, such as T central memory (Tcm) cells, neutrophils, mast cells, eosinophils, and gamma delta T cells (Tgd). Additionally, T helper (Th) cells, Th17 cells, Th2 cells, and effector memory T (Tem) cells were examined. This study also considered macrophages, immature dendritic cells (iDCs), natural killer (NK) cells, and dendritic cells (DCs). Follicular helper T (TFH) cells, plasmacytoid dendritic cells (pDCs), Th1 cells, and NK CD56dim cells were also analyzed. The evaluation included activated dendritic cells (aDCs), B cells, CD8^+^ T cells, T cells, regulatory T cells (Tregs), cytotoxic cells, and NK CD56 bright cells. Additionally, the Wilcoxon rank-sum test was used to evaluate the differences in infiltration levels between groups with high and low NCKAP1 expression. To investigate the immune characteristics across different NCKAP1 expression groups, we used the CIBERSORT method to assess NCKAP1 expression and the differential expression of 22 immune cell types within the TCGA dataset. We used the expression profile data from each sequenced sample to evaluate the relative expression of NCKAP1, aiming to determine the proportions of 24 distinct immune cell types. We subsequently investigated the relationships between these immune cell subpopulations and their gene expression patterns. To visualize these correlations, we used bar plots, box plots, and heatmaps, which were created via the R packages “ggpubr” and “corrplot”. We explored the relationships between various immune cells and NCKAP1 via Pearson’s correlation analysis in R, with visualizations generated via the “ggplot2” package and a significance threshold of *p* < 0.05.

### 5.7. Cell Culture and Samples

The human renal cell carcinoma cell lines 786O and 769P were obtained from our laboratory. The cells were cultured in RPMI-1640 medium (Gibco, Carlsbad, CA, USA) supplemented with 10% fetal bovine serum (FBS, HyClone, Logan, UT, USA) and 1% penicillin/streptomycin (Sigma-Aldrich, St. Louis, MO, USA) at 37 °C in a humidified 5% CO_2_ atmosphere. The tumor samples were collected in the Department of Urology at the Shanghai Public Health Clinical Center and originated from surgical procedures performed on patients with renal cell carcinoma. Ethical approval for the use of these human samples was obtained from the Institutional Review Board (IRB) (IRB Number: 2020-S060-01).

### 5.8. CCK8 Assay

Cell viability was assessed by the CCK-8 assay (Dojindo, Kumamoto, Japan). Briefly, cells were seeded in 96-well plates at a density of 5 × 10^3^ cells/well. After treatment, 10 μL of CCK-8 reagent was added to each well and incubated for 2 h at 37 °C. The absorbance was measured at 450 nm via a BTI-ELx808 microplate reader (Gene Company Limited, Hong Kong, China; Model# BTI-ELx808) equipped with a monochromator-based optical system. Each experiment was performed in triplicate and repeated three times independently.

### 5.9. Transwell Assay

Cell migration was evaluated using Transwell chambers (8 μm pore size; Corning, NY, USA). For the invasion and migration assays, the chambers were precoated with or without Matrigel (BD Biosciences, San Jose, CA, USA). The cells (1 × 10^5^ in serum-free medium) were seeded into the upper chamber, while the lower chamber contained complete medium supplemented with 10% FBS. After 48 h of incubation, the migrated cells on the lower membrane surface were fixed with 4% paraformaldehyde, stained with 0.1% crystal violet, and photographed under an inverted microscope (Olympus IX73, Tokyo, Japan). Cell numbers were quantified by a hemocytometer. All experiments were performed in triplicate and repeated at least three times independently.

### 5.10. Quantitative Real-Time PCR

Using TRIzol^®^ Reagent (TIANGEN, Beijing, China), complete RNA was isolated from both the tissue samples and the cellular material. The quality of the RNA was evaluated via a spectrophotometer, with OD260/280 ratios between 1.8 and 2.0 considered acceptable. Genomic DNA was extracted via the PrimerScriptTM RT Reagent Kit with gDNA Eraser (RR047A, Takara, Kyoto, Japan). This process involved incubation at 42 °C for 5 min, followed by an indefinite period at 4 °C. The PrimerScriptTM RT Reagent Kit with Thank you for your reminder. All highlighted parts have been added with the corresponding cities.gDNA Eraser (RR047A, Takara, Japan) was used for reverse transcription with the following protocol: 60 min at 37 °C, 5 s at 85 °C, and an indefinite hold at 4 °C. Subsequently, quantitative real-time PCR (qRT-PCR) was conducted via Takara SYBR^®^ Premix Ex Taq™ II (RB820A, Takara, Japan) on a Roche LightCycler^®^ 480 II instrument (Roche Diagnostics, Rotkreuz, Switzerland). The expression of NCKAP1 was normalized to that of 18S rRNA, which served as an internal control. For NCKAP1 and 18S rRNA, the following primer sequences were used: NCKAP1-F: TCCTAAATACTGACGCTACAGCA; NCKAP1-R: GCCTCCTTGCATTCTCTTATGTC; 18S rRNA-F: GGAGTATGGTTGCAAAGCTGA; and 18S rRNA-R: ATCTGTCAATCCTGTCCGTGT.

### 5.11. Immunoblots

Cell lysate proteins were separated by SDS–PAGE and transferred onto a nitrocellulose membrane with a 0.45 μm pore size. First, the membranes were blocked with 5% nonfat milk in PBST for 60 min. Subsequently, they were incubated overnight with primary antibodies at 4 °C, followed by a one-hour incubation with secondary antibodies at room temperature (25 °C). An automated chemiluminescence imaging system was used to measure the protein concentrations of HSP90, vimentin, snail, E-cadherin, mTOR, phosphorylated mTOR (P-mTOR), AKT, phosphorylated AKT (P-AKT), PI3K, and phosphorylated PI3K (P-PI3K). Proteins were visualized via an enhanced chemiluminescence (ECL) system (Santa Cruz Biotechnology, Santa Cruz, CA, USA).

### 5.12. Statistical Analysis

Data analysis was conducted via R software (version 3.6.3), which employs a range of statistical techniques. These included Kaplan–Meier (KM) survival analysis and log-rank tests, which were used to assess prognostic significance and evaluate differences in patient survival among various subgroups. For groups with a normal distribution, Student’s *t* test was employed, whereas the Wilcoxon test was used for variables that were not normally distributed. Spearman’s correlation analysis was used to evaluate relationships between variables. Statistical significance was set at *p* < 0.05.

## Figures and Tables

**Figure 1 ijms-26-02813-f001:**
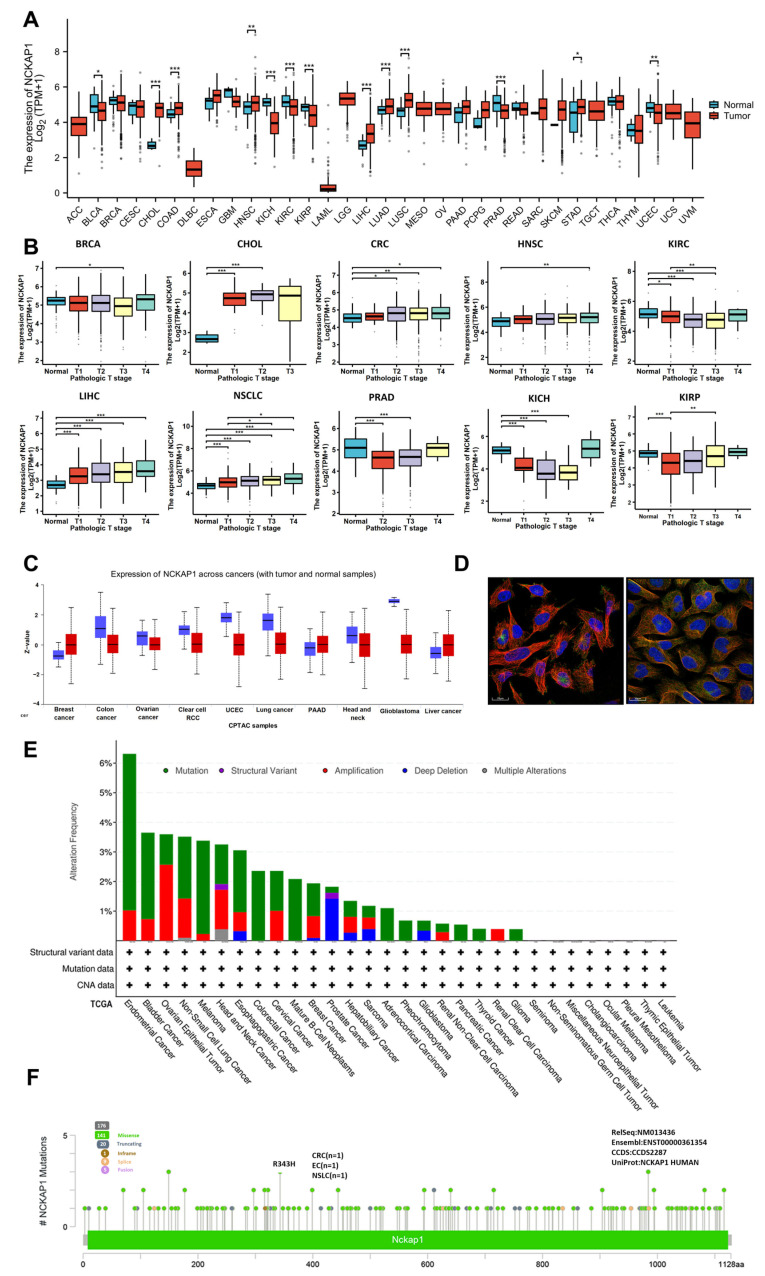
Nck-associated protein 1 (NCKAP1) is expressed across various tumor types and stages. (**A**) Visualization of NCKAP1 gene expression in 33 different tumor types and corresponding normal tissues from the TCGA datasets. (**B**) Analysis of NCKAP1 expression across stages I to IV in the TCGA datasets focused on BRCA, CHOL, CRC, HNSC, and KIRC. The expression levels are presented on a log2 scale (TPM + 1). Each point represents the expression level of NCKAP1 in an individual sample. (**C**) Comparative analysis of NCKAP1 protein expression between tumor and adjacent normal tissues across 10 cancer types via data from the UALCAN database. (**D**) Subcellular localization of NCKAP1 in tumor cells, as determined via the HPA database. Red fluorescence indicates microtubules, whereas green fluorescence indicates NCKAP1 distribution. (**E**,**F**) Genetic alteration assessment of NCKAP1 via cBioPortal. (**E**) Summary of alteration frequencies across various cancer types, detailing the spectrum of observed mutation types. (**F**) Detailed presentation of specific mutation types, their genomic locations, and the number of cases exhibiting genetic alterations in NCKAP1. * *p* < 0.05; ** *p* < 0.01; *** *p* < 0.001.

**Figure 2 ijms-26-02813-f002:**
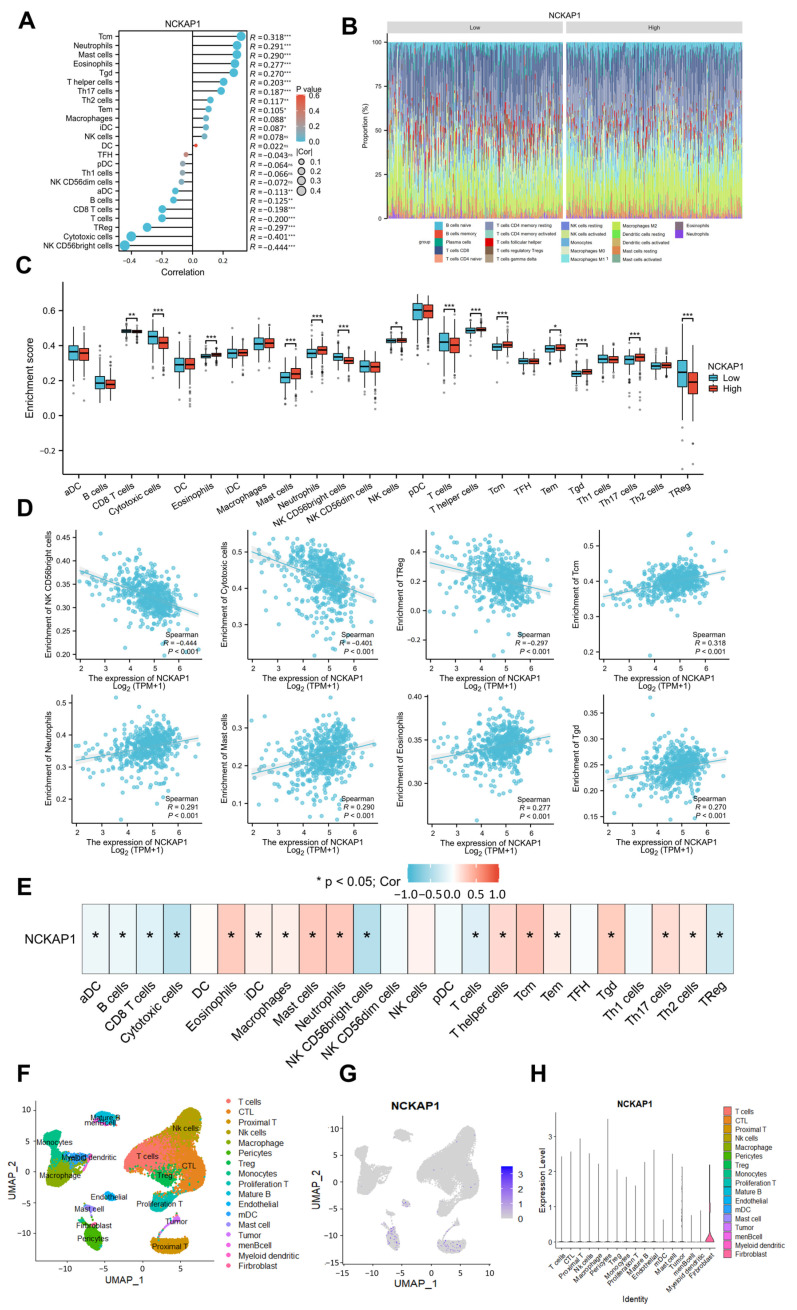
Analysis revealed that NCKAP1 expression is related to immune infiltration in pancancer cells. (**A**) Lollipop graph illustrating the correlation between NCKAP1 expression and 24 distinct immune cell types. The color intensity of the bubbles indicates the *p* value, whereas the size denotes the strength of the correlation. The terms on the left side indicate a negative correlation, and those on the right side indicate a positive correlation, both with *p* values less than 0.05. (**B**) Bar plot showing the relative proportions of 22 immune cell subsets in the groups with high and low NCKAP1 expression. The *X*-axis denotes the expression status of NCKAP1, and the *Y*-axis indicates the percentage of 22 different immune cell types. The percentages were calculated individually for each gene expression series. (**C**) Box plot comparing the infiltration levels of 24 immune cells between the groups with high and low NCKAP1 expression. (**D**) NK CD56 bright cells (*p* < 0.001, r = 0.177), cytotoxic cells (*p* < 0.001, r = 0.177), and Tregs (*p* < 0.001, r = 0.177) were negatively associated with NCKAP1, whereas Tcm (*p* < 0.001, r = 0.318), neutrophils (*p* < 0.001, r = 0.291), mast cells (*p* < 0.001, r = 0.290), eosinophils (*p* < 0.001, r = 0.277), and Tgd (*p* < 0.001, r = 0.270) were positively associated with NCKAP1. (**E**) Spearman’s correlation of NCKAP1 with 24 types of immune cells. (**F**) UMAP plot illustrating the annotation and color codes for 17 cell types within the ccRCC ecosystem derived from single-cell transcriptional analyses of NCKAP1. (**G**) This UMAP plot illustrates the distribution of NCKAP1 expression within renal cancer samples. (**H**) Plots showing the expression levels of NCKAP1 across 17 immune cell types, with fibroblasts exhibiting the highest expression. * *p* < 0.05; ** *p* < 0.01; *** *p* < 0.001.

**Figure 3 ijms-26-02813-f003:**
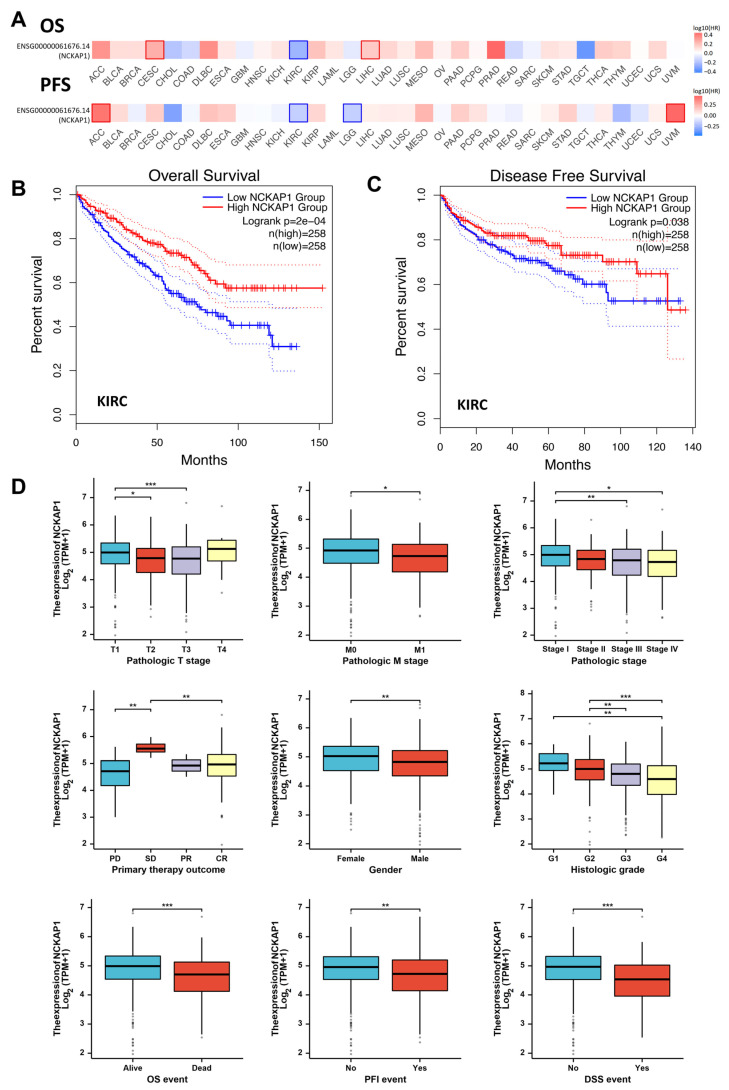
Survival analysis of NCKAP1 across various cancers via GEPIA2. (**A**) Heatmap depicting OS and DFS, illustrating that high NCKAP1 expression is associated with poor prognosis in ACC, CESC, and LIHC. Conversely, high expression correlated with better prognosis in patients with KIRC, TGCT, and LGG. (**B**,**C**) KM curves for OS and DFS demonstrate that high NCKAP1 expression is associated with improved prognosis. The solid red line represents the estimated survival rate for samples with high expression of NCKAP1, while the solid blue line represents the estimated survival rate for samples with low expression of NCKAP1. The dotted red and blue lines indicate the 95% confidence intervals for these estimates. (**D**) Box plot depicting the associations between NCKAP1 expression and various clinicopathological parameters in renal cancer patients. Each point represents the expression level of NCKAP1 in an individual sample. * *p* < 0.05; ** *p* < 0.01; *** *p* < 0.001.

**Figure 4 ijms-26-02813-f004:**
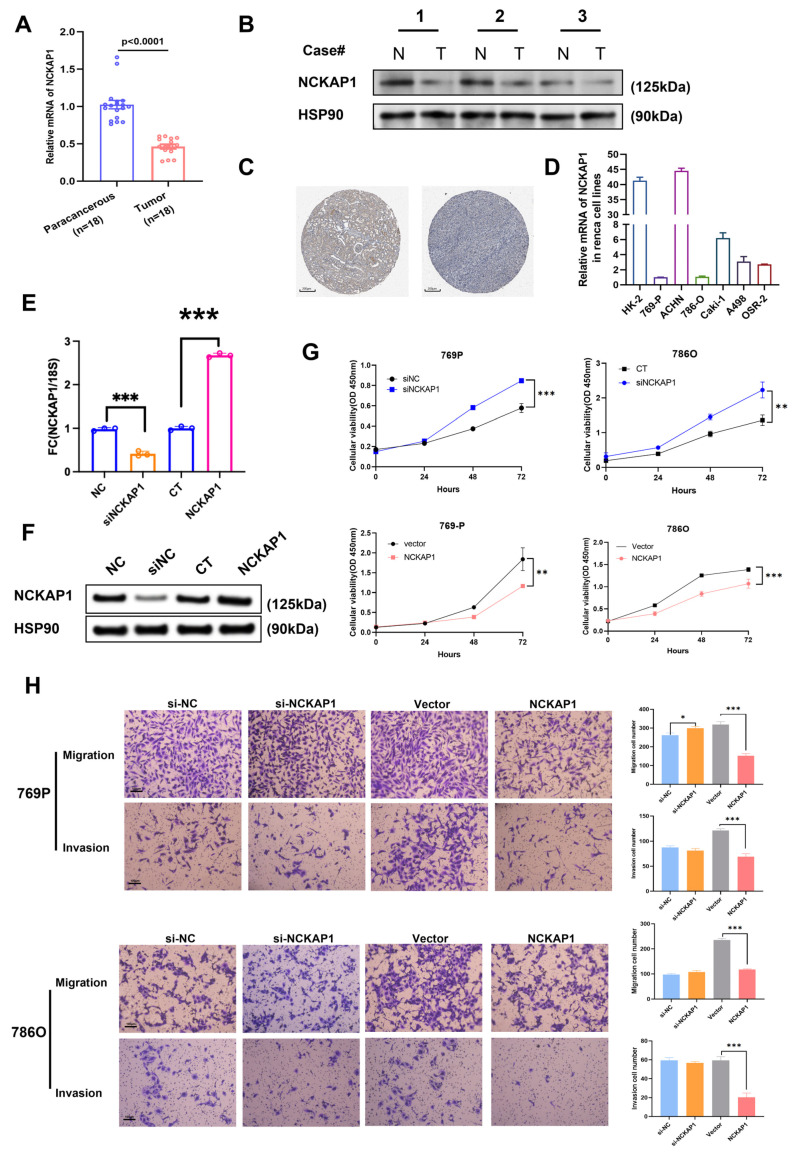
NCKAP1 expression influences renal cancer cell proliferation, migration, and invasion. (**A**) NCKAP1 mRNA expression levels in precancerous and tumor tissues from patients with renal cancer. (**B**) Western blot analysis revealed the protein expression levels of NCKAP1 in several pairs of renal cancer and adjacent normal tissues. (**C**) Comparison of the protein expression of NCKAP1 between kidney and renal cancer tissues via the Human Protein Atlas. (**D**) The expression levels of NCKAP1 across various renal cancer cell lines. (**E**) NCKAP1 expression was assessed via RT-PCR in 786O and 769P cells after transfection with siRNA or OE-NCKAP1. (**F**) NCKAP1 expression was assessed via WB in 786O and 769P cells after transfection with siRNA or OE-NCKAP1. (**G**) Representative data from the CCK8 assay performed on 769P and 786O cells with low and high expression of NCKAP1. (**H**) Representative data from Transwell migration and Matrigel invasion assays performed in 769P and 786O cell lines exhibiting varying levels of NCKAP1 expression. * *p* < 0.05; ** *p* < 0.01; *** *p* < 0.001.

**Figure 5 ijms-26-02813-f005:**
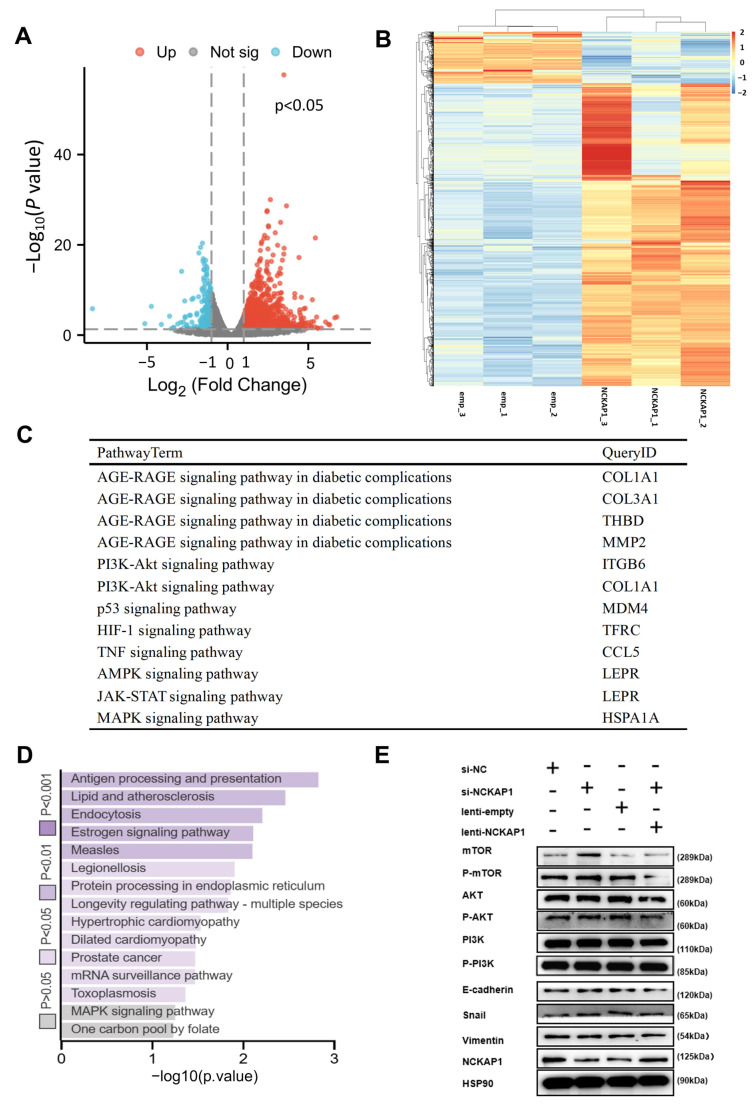
Comprehensive analysis of RNA sequencing data revealed the impact of NCKAP1 on gene expression and signaling pathways in renal cancer cells. (**A**) Volcano plot of DEGs in NCKAP1-overexpressing 769P cells compared with control cells. (**B**) Heatmap illustrating the differential expression signatures. (**C**) Cancer-pathway-related KEGG enrichment analysis of the 30 genes whose expression was most significantly upregulated or downregulated. (**D**) List of KEGG pathways enriched with the top 30 upregulated and downregulated genes. (**E**) Western blot analysis of the regulatory effect of NCKAP1 on the PI3K/AKT/mTOR and EMT signaling pathways.

**Table 1 ijms-26-02813-t001:** Analyses of NCKAP1 expression in renal cancer patients and the clinical features of patients with KIRC, KIRP, and KICH in the TCGA datasets (bold denotes statistical significance (*p* value ≤ 0.05)).

Characteristics	KIRC		*p* Value	KIRP		*p* Value	KICH		*p* Value
	NCKAP1^Low^	NCKAP1^High^		NCKAP1^Low^	NCKAP1^High^		NCKAP1^Low^	NCKAP1^High^	
n	270	271		145	146		32	33	
Gender, n (%)			**0.016**			**<0.001**			0.362
Female	80 (14.8%)	107 (19.8%)		19 (6.5%)	58 (19.9%)		11 (16.9%)	15 (23.1%)	
Male	190 (35.1%)	164 (30.3%)		126 (43.3%)	88 (30.2%)		21 (32.3%)	18 (27.7%)	
Pathologic T stage, n (%)			**0.036**			**0.013**			0.535
T1 and T2	163 (30.1%)	187 (34.6%)		121 (41.9%)	106 (36.7%)		21 (32.3%)	24 (36.9%)	
T3 and T4	107 (19.8%)	84 (15.5%)		22 (7.6%)	40 (13.8%)		11 (16.9%)	9 (13.8%)	
Pathologic N stage, n (%)			0.540			**0.020**			0.198
N0	117 (45.3%)	125 (48.4%)		28 (35.9%)	22 (28.2%)		24 (54.5%)	15 (34.1%)	
N1	9 (3.5%)	7 (2.7%)		8 (10.3%)	20 (25.6%)		1 (2.3%)	4 (9.1%)	
Pathologic M stage, n (%)			0.051			0.605			1.000
M0	204 (40.2%)	225 (44.3%)		46 (44.2%)	49 (47.1%)		21 (58.3%)	13 (36.1%)	
M1	47 (9.3%)	32 (6.3%)		3 (2.9%)	6 (5.8%)		1 (2.8%)	1 (2.8%)	
Pathologic stage, n (%)			**0.006**			**0.022**			0.535
Stage I and Stage II	150 (27.9%)	182 (33.8%)		101 (38.7%)	93 (35.6%)		21 (32.3%)	24 (36.9%)	
Stage III and Stage IV	118 (21.9%)	88 (16.4%)		24 (9.2%)	43 (16.5%)		11 (16.9%)	9 (13.8%)	
Histologic grade, n (%)			**<0.001**						
G1 and G2	105 (19.7%)	145 (27.2%)							
G3 and G4	162 (30.4%)	121 (22.7%)							
OS event, n (%)			**<0.001**			0.529			0.096
Alive	161 (29.8%)	205 (37.9%)		125 (43%)	122 (41.9%)		30 (46.2%)	25 (38.5%)	
Dead	109 (20.1%)	66 (12.2%)		20 (6.9%)	24 (8.2%)		2 (3.1%)	8 (12.3%)	
DSS event, n (%)			**<0.001**			0.125			0.066
Yes	75 (14.2%)	34 (6.4%)		10 (3.5%)	18 (6.3%)		1 (1.5%)	7 (10.8%)	
No	189 (35.7%)	232 (43.8%)		132 (46%)	127 (44.3%)		31 (47.7%)	26 (40%)	
PFI event, n (%)			**0.002**			0.115			0.063
Yes	97 (17.9%)	65 (12%)		24 (8.2%)	35 (12%)		3 (4.6%)	9 (13.8%)	
No	173 (32%)	206 (38.1%)		121 (41.6%)	111 (38.1%)		29 (44.6%)	24 (36.9%)	

## Data Availability

The data used in this article are mentioned in the methods section.

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
