# Peer review of "NCKAP1 Inhibits the Progression of Renal Carcinoma via Modulating Immune Responses and the PI3K/AKT/mTOR Signaling Pathway"

_ijms, 2025, doi:10.3390/ijms26062813_

Round 1

Reviewer 1 Report

Comments and Suggestions for Authors

Reviewer comments:

Title: NCKAP1 Inhibits the Progression of Renal Carcinoma via Modulating Immune Responses and the PI3K/AKT/mTOR Signaling Pathway

The manuscript by Zhang et. al., describes about the Nck-associated protein1 (NCKAP1) which is found to be deregulated in various cancers and influences the malignancy levels. The authors have revealed the function and genetic variations of KCKAP1 in renal cancer by using various web-based tools. They found that the expression levels of NCKAP1 are significantly reduced in renal cancer patients and correlated with immune cell infiltration. They concluded that NCKAP1 suppresses the cancer growth via PI3K/AKT/mTOR Signaling Pathway and can be a potential marker for targeting renal cancer.

The manuscript is very well written, but some points need to be addressed.

  1. There are other literatures by Chen et al., 2022 (10.3389/fgene.2022.764957), Liang et al., 2024 (10.62347/UKQB2042) and others, which also describes similar work and results? How your work is significantly different from these?
  2. There is very limited literature about KIRP cancer, why the authors have not focused on this cancer with relation to NCKAP1 protein?
  3. Figure 1A: there seems no significant difference between tumor and normal groups in KIRC. If there are any, please show the results in boxplot.
  4. Page 17, para 2, line 7: the line can be modified as “expression with log2 fold change cutoff of ±1”. Please mention the two groups used for preparing volcano plot in the text as well as in legends of figure 5.
  5. Please add a conclusion section.

Author Response

Reviewer comments:

The manuscript by Zhang et. al., describes about the Nck-associated protein1 (NCKAP1) which is found to be deregulated in various cancers and influences the malignancy levels. The authors have revealed the function and genetic variations of KCKAP1 in renal cancer by using various web-based tools. They found that the expression levels of NCKAP1 are significantly reduced in renal cancer patients and correlated with immune cell infiltration. They concluded that NCKAP1 suppresses the cancer growth via PI3K/AKT/mTOR Signaling Pathway and can be a potential marker for targeting renal cancer.

Response: Thank you for your good comments.

The manuscript is very well written, but some points need to be addressed.

1.  There are other literatures by Chen et al., 2022 (10.3389/fgene.2022.764957), Liang et al., 2024 (10.62347/UKQB2042) and others, which also describes similar work and results? How your work is significantly different from these?

Response: Thank you for your suggestion. We have cited the references (Ref 14: Chen et al., 2022 (10.3389/fgene.2022.764957), Ref 21: Liang et al., 2024 (10.62347/UKQB2042)) and added “NCKAP1 is also deregulated in renal cancer and is associated with patient outcomes [14, 21]” in the discussion of the revised manuscript.

 2.  There is very limited literature about KIRP cancer, why the authors have not focused on this cancer with relation to NCKAP1 protein?

Response: Thank you for your good suggestions. We have analyzed the expression of NCKAP1 in KIRC and added the clinical and prognostic data related to NCKAP1 in KIRC and KIRP in Table 1 and Supplemental Figure S1, and added “To evaluate the clinical significance of NCKAP1, we analyzed the relationships between NCKAP1 expression and clinical features in 897 patients with renal cancer including KIRC (n = 541), KIRP (n = 291) and KICH (n = 65) from the TCGA datasets (Table 1). Reduced NCKAP1 expression in KIRC was significantly correlated with more advanced stages of pathological T stage (p = 0.035), overall pathological stage (p = 0.006), and higher histological grade (p = 0.0004). We also demonstrated that NCKAP1 expression significantly affected OS (p < 0.0001), DSS (p < 0.0001), and PFI (p = 0.002) in patients with KIRC, although NCKAP1 expression was not significantly correlated with pathological grade or clinical prognosis in KIRP or KICH patients (Table 1 and Supplemental Figure S1B)” in the results of the revised manuscript.

3. Figure 1A: there seems no significant difference between tumor and normal groups in KIRC. If there are any, please show the results in boxplot.

Response: Thank you for your reminding. We have replaced Figure 1A for the expression of NCKAP1 in tumor and adjacent normal tissues of KIRC and added the Supplemental Figure S1A which clearly illustrate the differences in NCKAP1 expression between the two groups.

4. Page 17, para 2, line 7: the line can be modified as “expression with log2 fold change cutoff of ±1”. Please mention the two groups used for preparing volcano plot in the text as well as in legends of figure 5.

Response: Thank you for your good suggestions. We have modified as “a log2 fold change threshold of ± 1” and added “in NCKAP1-overexpressing cells compared with control cells” in the results. We have also replaced the legend of Figure 5A.

5. Please add a conclusion section.

Response: Thank you for your good suggestions. We have added “In this study, we demonstrated that NCKAP1 inhibited the growth and progression of renal cancer cells via the PI3K/AKT/mTOR signaling pathway, providing a molecular basis for its potential as a therapeutic target. Furthermore, CIBERSORT and single-cell RNA sequencing data revealed that NCKAP1 had significant immunomodulatory effects. Considering the complexities revealed by our pancancer analysis, future research should focus on elucidating the mechanistic role of NCKAP1 in various cancers. Additionally, more clinical data, particularly from online databases and animal experiments, should be used to validate these findings. We should also investigate the specific pathways influenced by NCKAP1 and evaluate the impact of its modulation on clinical outcomes. These studies could pave the way for personalized therapeutic strategies, particularly for renal cancer, where the impact of NCKAP1 is more pronounced. This study advances our understanding of the roles of NCKAP1 in cancer and opens new avenues for personalized treatment strategies, potentially transforming clinical practice” in the conclusion of the revised manuscript.

Reviewer 2 Report

Comments and Suggestions for Authors

In this paper, Zhang et al. performed a comprehensive analysis of NCKAP1 in cancer. They compared the expression and genetic alterations of NCKAP1 across various cancer cell lines and explored the correlation between NCKAP1 expression and changes in immune cells within the tumor microenvironment. Furthermore, the authors studied the significance of NCKAP1 as a prognostic biomarker in various renal cancer types and demonstrated that its effects are mediated through the PI3K/AKT/mTOR pathway. Altogether, the findings from this study strongly suggest that NCKAP1 is a potential tumor suppressor in renal carcinoma. The bioinformatics data analysis is thorough, utilizing appropriate statistical tests. The authors also supported their findings with knockdown and overexpression studies in cell line models. In my opinion, this study is well-conducted with rigorous data analysis, and I do not have any specific comments or suggestions for the authors. I recommend this study for publication.

Author Response

Review 2

In this paper, Zhang et al. performed a comprehensive analysis of NCKAP1 in cancer. They compared the expression and genetic alterations of NCKAP1 across various cancer cell lines and explored the correlation between NCKAP1 expression and changes in immune cells within the tumor microenvironment. Furthermore, the authors studied the significance of NCKAP1 as a prognostic biomarker in various renal cancer types and demonstrated that its effects are mediated through the PI3K/AKT/mTOR pathway. Altogether, the findings from this study strongly suggest that NCKAP1 is a potential tumor suppressor in renal carcinoma. The bioinformatics data analysis is thorough, utilizing appropriate statistical tests. The authors also supported their findings with knockdown and overexpression studies in cell line models. In my opinion, this study is well-conducted with rigorous data analysis, and I do not have any specific comments or suggestions for the authors. I recommend this study for publication.

Response: Thank you for your good comments.
